# Analysis of the Effect of Magnetic Field on Solidification of Stainless Steel in Laser Surface Processing and Additive Manufacturing

**Svetlana A. Gruzd** [1,*], **Stepan L. Lomaev** [2], **Nikolay N. Simakov** [1], **Georgii A. Gordeev** [1], **Anton S. Bychkov** [3], **Artem A. Gapeev** [3], **Elena B. Cherepetskaya** [3], **Mikhail D. Krivilyov** [1,2,*] and **Ivan A. Ivanov** [4,5]

1   Laboratory of Condensed Matter Physics, Department of Physics, Udmurt State University, Universitetskaya Str. 1, 426034 Izhevsk, Russia
2   Udmurt Federal Research Center of the Ural Branch of RAS, Baramzina Str. 34, 426067 Izhevsk, Russia
3   Laboratory of Laser-Ultrasonic Diagnostics, The National University of Science and Technology MISiS, Leninskiy Prospect 4, 119991 Moscow, Russia
4   JSC "NPO "TSNIITMASH", Sharikopodshipnikovskaya Str. 4, 115088 Moscow, Russia
5   LLC "Rusatom-Additive Technology" Industrial Integrator the SC Rosatom, Kashirskoye Highway 49, 115409 Moscow, Russia
*   Correspondence: lilyna@mail.ru (S.A.G.); mk@udsu.ru (M.D.K.)

**Abstract:** The problem of surface processing and microstructure refinement in stainless steels in laser surface processing in the presence of an external magnetic field has been studied experimentally and theoretically. The effect of both alternating and permanent magnetic fields is discussed. The experimental part includes microstructure assessment of a thin stainless plate annealed by a quasi-continuous laser in the presence of an electromagnetic acoustic transducer. Complementary analytical calculus and numerical simulations of complex transport phenomena in the melting zone are performed. Based on the received data, the effect of the electromagnetic field on the molten zone under laser melting conditions is evaluated and quantified. The obtained results are relevant to laser surface hardening and additive manufacturing.

**Keywords:** laser processing; stainless steels; electromagnetic field; multiphase flow; solidification; microstructure control; multiphysics approach; numerical simulations

## 1. Introduction

Improvement of mechanical strength characteristics of manufactured metal products is an important goal directly related to alloy microstructure. One of the industrially implemented methods for resolving this technological problem is laser annealing the metal surfaces [1]. The advantage of this method is primarily ensured by local heat treatment within a short time. As a result, high temperature gradients, cooling rates and solidification velocities are generated in the local volume of the molten material. Hence, microstructure refinement and mechanical hardening occur.

In [2–5], the authors suggest various approaches for adjustment processing parameters in laser techniques, including selective laser manufacturing (SLM) to enhance the performance characteristics of metal parts. The velocity of hydrodynamic flow is one of the most important physical parameters in the molten pool formed by a laser source. Starting from applications in the heavy engineering industry and concluding by levitation experiments in microgravity, control and correction of convective flow in liquid metals is widely performed using external electromagnetic (EM) fields [6–12]. For instance, the alternating EM field leads to the additional force affecting the fluid flow in the molten pool [6]. The EM stirring induced by the Lorentz force can slow down or accelerate the flow dependent on the modulation of the magnetic field [7]. This effect corresponds to the intensive thermocapillary convection in SLM, which yields even more complex situations from the position of its

engineering control. The permanent magnetic field leads to deceleration of the convective flow in the melt owing to the Hartmann effect known from the literature [13].

There are few applications where the EM field is utilized in various technological processes. Another example is the flow control by the magnetic field in the molten pool during laser welding of thick austenite sheets. The electromagnetic field improves the quality of the weld and increases its strength [13]. At the same time, the melt is additionally cleaned from oxides and gas bubbles under the electromagnetic influence. High sensitivity of aluminum alloys in solidification under the magnetic field has been reported [14]. According to the authors [14], the resulting microstructure is altered by a strong thermoelectric effect. The relevant magnetic impact is reported in [15], where the different magnetic field strength leads to fragmentation of dendrites and their shift in the direction perpendicular to the magnetic field. The authors indicate that the magnetic field results in instability of the solid–liquid interface. Thus, intensification of chemical segregation coupled with an expansion of the mushy zone proceeds.

A series of research is devoted to the magnetic field impact on laser welding of Al alloys [16–19]. The localized molten zone is characterized by inhomogeneous distribution of alloying elements and non-metallic inclusions in the seam. Therefore, reinforcement of convection helps with chemical homogeneity and reduces the weld bead thanks to better flow stability in the melted pool.

High temperature gradients in the localized molten zone annealed by a laser appear due to ultra-fast heating and subsequent cooling. Such conditions promote thermoelectric currents known as the Seebeck effect in the literature. For example, in [20], the impact of the magnetic field is related to the Seebeck effect in the $Al_{90}Si_{10}$ melt. Variations of the convective heat transport changes the mass transport. Consequently, the kinetics of solidification and grain formation is also affected. Similar results have been delivered for Al–10%Si–Mg [21] in connection with the Seebeck effect in SLM. The experimental tests with the superimposed magnetic field showed that the primary dendritic space decreases as well as the volume fraction of columnar grains. In the same reference, the results of mechanical tests are provided and the meaningful impact is confirmed. The abovementioned discussion evidences that the high density of the heat flux, small time period of presence of the molten pool, and different auxiliary effects result in complex mutual coupling of various physical phenomena. Hence, no simple empirical model connects this problem's processing parameters and resulting characteristics.

Steel is the most common alloy in metallurgical production. In the additive manufacturing domain, stainless steels similar to the austenite 316L steel are broadly employed [22,23]. Thus, a question if the EM field can modify the resulting properties of SLM-produced parts is under active consideration. Furthermore, it is essential to establish whether any significant effect on microstructure can be achieved in steels like it was established in Al alloys under the EM field. Owing to a substantial difference between Fe and Al alloys in the mechanisms of microstructure formation, the conclusions obtained for Al alloys are not directly applicable to steels.

The present work considers the effect of the magnetic field on heat and mass transfer coupled with solidification of the molten pool in laser surface remelting of stainless steel. The motivation is based on the fact that this problem is of technological interest; however, it is not well studied in the context of additive manufacturing. The scientific merit of this paper is a comprehensive analysis of the impact on microstructure under different levels of the EM field. To that end, the experimental data on grain statistics are confronted with the results of numerical simulations.

## 2. Experimental Methodology and Microstructure Assessment

In the previous work [5], the authors considered the effect of laser-induced ultrasound on the microstructure of laser-processed stainless steel. Significant grain refinement was confirmed experimentally in steel plates processed by the sub-MHz intensity-modulated pulsed laser. Here, the impact of the alternating EM field on solidification of AISI 321H

stainless steel under localized pulsed laser exposure is studied experimentally on a plate 20 mm × 20 mm × 5 mm in size. The surface of the plate is exposed to focused laser pulses (wavelength 1.06 μm, 160 mm focal length, pulse energy 45.1 mJ, laser spot diameter ~0.2 mm and pulse repetition frequency 2 Hz). The long-term average optical power was ~90 mW, and the sample material had enough time to solidify after each laser pulse. An EM acoustic transducer (EMAT, model ULTRASTIR, ULTRAKRAFT, Cherepovets, Russia) [24] was additionally applied to the plate. The EMAT was positioned coaxially with the laser beam, so that the magnetic field was perpendicular to the surface of the sample (Figure 1).

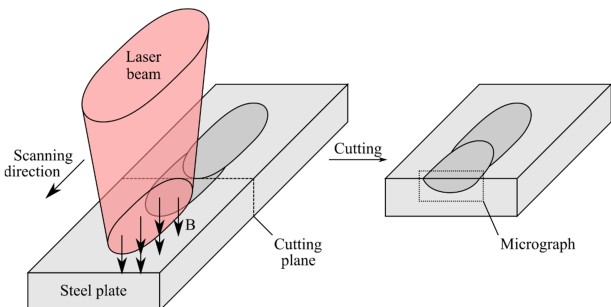

**Figure 1.** Scheme of the experimental setup.

The Nd:YAG laser operated in the free-running mode generating envelopes of ~150 μs long filled with irregularly spaced spikes of ~1 μs long. The Fourier spectrum of laser pulses had a prominent low-frequency component from the envelope and a high-frequency component stretching from 250 kHz to 1 MHz from the spikes. Although the peak power of the spikes exceeded 1 kW, the power of the envelope rose up to ~400 W within the first ~20 μs of the pulse and decayed for the rest of the pulse. The time dependence of the laser optical power is shown in Figure 2. The diameter of the working area of the EMAT was 30 mm, the center frequency was 100 kHz and the average power was 10 W. In Figure 2, after the end of the laser pulse, regular pulses are visible—these are cross-talk created by the EMAT on the electronics of the photodiode used to register the laser pulse.

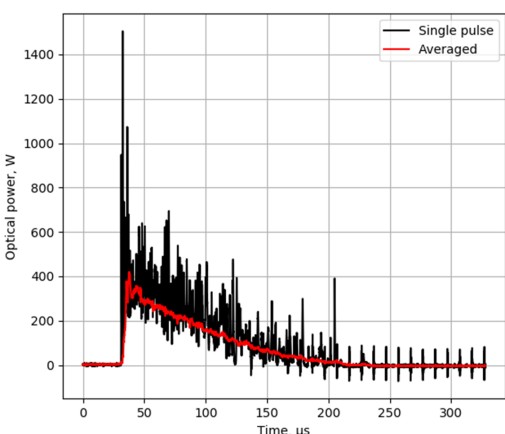

**Figure 2.** The time dependence of the laser optical power. The black line represents a single pulse. The red line is averaged over 404 laser pulses.

After each laser pulse the plate was shifted horizontally by a distance of 50 μm so that the next laser spot overlapped with the previous one. Thus, a laser-melted line segment was formed at the top surface of the plate. Then, the plate was cut in the plane perpendicular to this line segment, and the microstructure of the cross-section was analyzed.

The experimental program included two stages: (1) the laser exposure was performed with the EMAT turned off and (2) with the EMAT turned on. At each stage, the effect of laser remelting of the line segment was studied—up to 5 passes of pulsed laser irradiation

were carried out along the same line segment. The resulting microstructures with EMAT turned off are shown in Figure 3, and those with EMAT turned on are shown in Figure 4. The coloring corresponds to the areas of micrograins, and the color bars are presented in Figure 5 above the histograms. The variation of melt pool depth in Figures 3 and 4 occurs due to the variability of the position of the cutting plane in metallographic processing of the samples. The separate slices may have a different distance to the center of the laser spot. Hence, the depth of the remelted zone in the micrographs is slightly different.

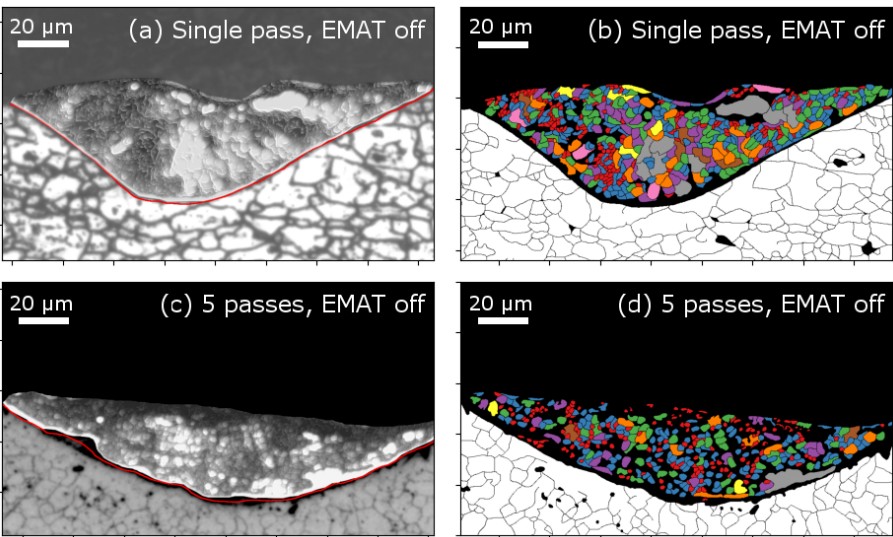

**Figure 3.** Metallographic images of microstructure of the AISI 321H stainless steel plate after pulsed laser exposure with EMAT turned off: (**a**) single laser pass and (**c**) five laser remelting passes. For the purpose of statistical analysis, the micrographs are computationally segmented and the grains are colored according to their size in (**b**,**d**) correspondingly. The same colors are used in Figure 5 (at the top of each histogram) in order to quantify the grain size intervals.

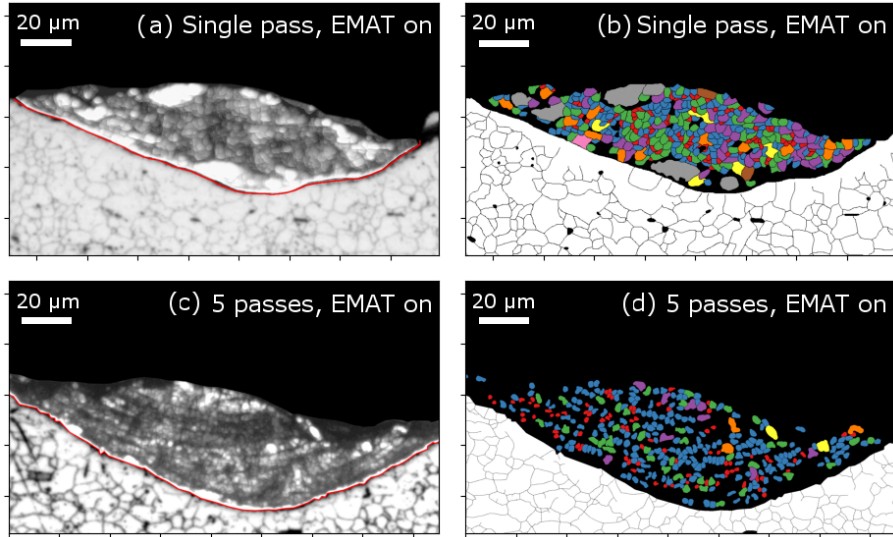

**Figure 4.** Metallographic images of microstructure of the AISI 321H stainless steel plate after pulsed laser exposure with EMAT turned on: (**a**) single laser pass and (**c**) five laser remelting passes. For the purpose of statistical analysis, the micrographs are computationally segmented and the grains are colored according to their size in (**b**,**d**) correspondingly. The same colors are used in Figure 5 (at the top of each histogram) in order to quantify the grain size intervals.

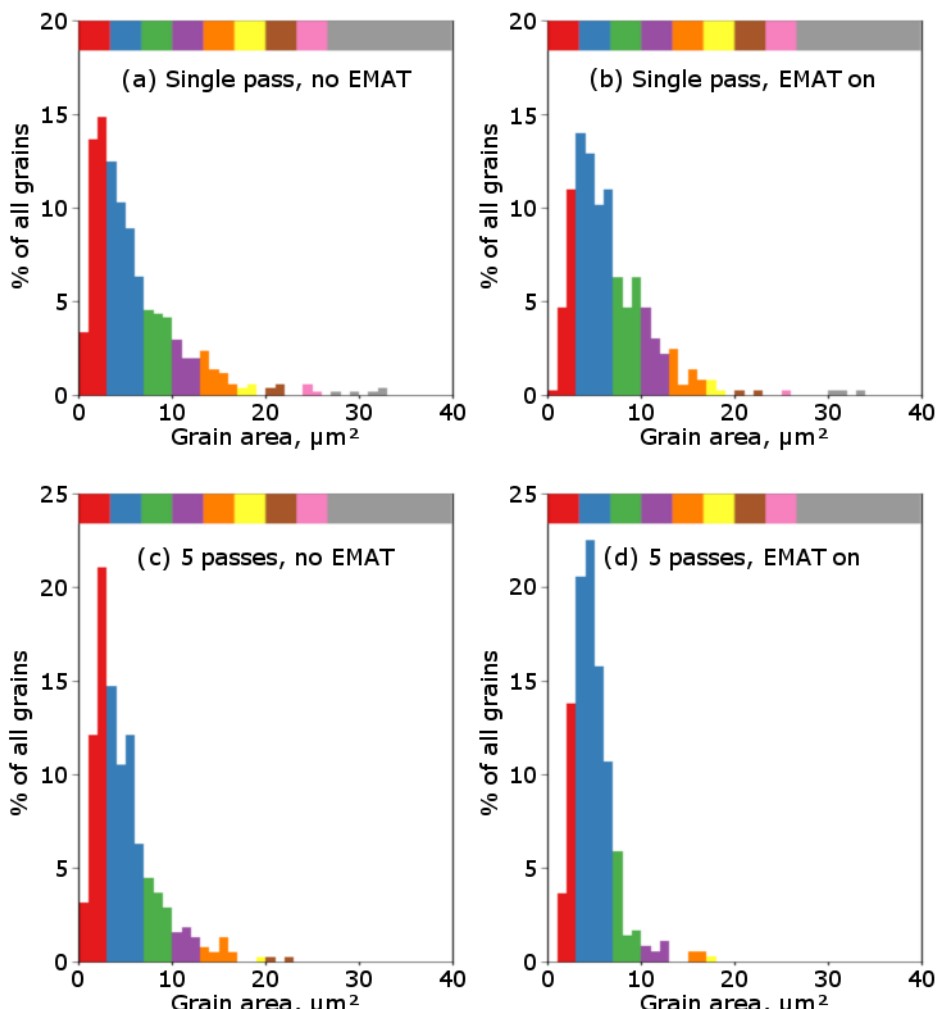

**Figure 5.** The distributions of micrograin sizes by their areas, (**a**) EMAT off, single pass; (**b**) EMAT on, single pass; (**c**) EMAT off, 5 passes; and (**d**) EMAT on, 5 passes. The same colors are used in Figures 3 and 4, where the grains are colored according to their size.

In Figure 3b,d and Figure 4b,d, the black regions correspond to the dark regions in sub-figures (a) and (c), where no white spots were identified. The total area of the dark regions increased with the number of laser passes. For a single pass, the total area of the dark regions was about 546 $\mu m^2$ (11% of the total melted area) with EMAT turned off and 581 $\mu m^2$ (15% of the total melted area) with EMAT turned on. After five subsequent passes, the total area of dark regions was about 1379 $\mu m^2$ (32% of the total melted area) with EMAT turned off and 2113 $\mu m^2$ (45% of the total melted area) with EMAT turned on. On the one hand, these regions could be filled with ultrafine grains. Since the authors were unable to identify them and therefore could not reliably measure their sizes, these dark areas were not taken into account when calculating the histograms. On the other hand, the dark regions could be attributed to the reduction in corrosion resistance due to remelting of small grains, which explains the increase in the area of the dark regions with the number of passes.

The micrograin sizes were analyzed for all the processing modes. The results of quantitative analysis for the micrograin sizes are presented in Figure 5 as histograms of the distribution of micrograins over their area. According to the experimental results, there are certain differences in the microstructure of the plate processed with and without EMAT. A small/moderate change in statistical characteristics was detected:

(1) Comparison of the histograms in Figure 5a–d, respectively, shows that the microstructure under the effect of magnetic fields is more homogeneous, and the spread in grain

sizes is smaller with EMAT turned on than with EMAT turned off. The fraction of grains with an area of less than 2 μm² or more than 15 μm² become smaller. Presumably, the reason for this is alignment of the flow conditions near the solidification front under the influence of an electromagnetic field. Random oscillations of the convective currents from uneven laser exposure are excluded that impacts on the spatial distribution of chemical components and microsegregation.

(2)　The initial microstructure was characterized by an average grain area of ~30 μm² and a median grain area of ~20 μm². With a single laser pass, no statistically significant differences between the modes with EMAT turned off/on were detected: the average areas were 6.5 and 7.4 μm², respectively, while the median areas were 4.5 and 5.6 μm², respectively.

(3)　With five laser passes, the effect of magnetic fields leads to reduction of the number of large grains. A shift to the left (toward smaller grain areas) is clearly visible in the histograms in Figure 5c,d, respectively, when exposed to an EM field.

Theoretical analysis of underlying phenomena during intensive plastic deformation of polycrystalline materials was performed in [25–27]. These studies showed that a transition from elastic to plastic deformation results in a different mechanism of plastic deformation. First, the planes of easy sliding are involved in deformation, then sliding occurs along the remained planes. At the latest stage, the mechanism of intergranular sliding and rotation is realized. At this stage, the homogeneous grain structure is preferable for better plasticity of the material. Therefore, the revealed in the present study change in the distribution of grain size potentially leads to a higher strain-to-fracture limit.

### 3. Relevant Physical Phenomena and Quantitative Estimation of the EM Field Effect

For proper analysis of the effects of an imposed external EM field on solidification theoretical analysis of physical processes in the molten pool is required. Following [6–12], one of the major paths of such impact is given by variation of the convective flow velocity.

The effect of the permanent magnetic field on hydrodynamics in a conductive liquid medium is usually described by the Hartmann number:

$$Ha = BR_b\sqrt{\frac{\sigma_{el}}{\eta_f}},$$

where $R_b$ is the characteristic length of the molten pool, $\eta_f$ is viscosity of the melt, $B$ is the magnetic flux density and $\sigma_{el}$ is the electrical conductivity. This number characterizes a ratio between the Lorentz force density and other dissipative (i.e., viscous) forces. The experimental data [28] evidence that the considered effect becomes significant at the level of Ha~100 and above. In the problem under consideration, these values are achieved at a magnetic flux density above $B$~10 T. Therefore, in the absence of electric currents without any additional voltage sources and with only internal thermoelectric voltage, the magnetic field with $B < 10$ T does not provide any significant impact on the melt flow in the molten pool. This conclusion was confirmed experimentally as discussed in what follows.

The Seebeck effect has to be accounted for quantitatively in connection to the considered coupling between the magnetic field and fluid flow. In [20,21,29], the effect of thermoelectric currents on microstructure formation was studied. In the present analysis of this phenomenon, the evaluation of the electric current density between the hot and cold subdomains is performed for the molten pool. The current density decays in time during laser heating of the laser annealed zone according to

$$j = \nabla T s \sigma_{el} e^{-\frac{t}{t_0}}, \tag{1}$$

where $\nabla T$ is the temperature gradient, $S$ is the Seebeck coefficient with its values determined for stainless steel in [30,31] and $t_0$ is the relaxation time. The performed calculations yield the estimate of $t_0$ of the order of

$$t_0 = \frac{\varepsilon_{0si}}{\sigma_{el}} \approx 10^{-18} \text{ s}. \tag{2}$$

The calculated relaxation time is sufficiently smaller than the characteristic time of temperature variation in the molten pool. Thus, the whole annealed sample is considered as a system where the current density is generated by the inhomogeneous thermal field in the vicinity of the molten pool almost instantaneously, i.e., with negligible relaxation. The governing impact on the system is given by the thermoelectric current that is generated owing to the transient temperature distribution under the laser beam moving with the velocity of $v_L$. The absolute magnitude of the thermoelectric current for stainless steel is then estimated as

$$j = \nabla^2 T S \varepsilon_{0si} v_L \approx 10^{-7} \frac{\text{A}}{\text{m}^2}. \tag{3}$$

The evaluated current density is much smaller than the Lorenz current induced by the transient magnetic field. In this connection, the thermoelectric term may be omitted in the current density equation if stainless steels are considered. In Al alloys, the Seebeck effect is higher because the Seebeck coefficient is larger in comparison to stainless steels. According to [32,33], the Seebeck coefficient is $10^{-5}$ B/K in Al alloys and about $10^{-7}$ B/K in steels [32,33].

The effect of the alternating EM field on hydrodynamics was discussed by us in earlier research [6,7]. In these papers, the characteristic Lo number was introduced equal to the ratio of the Lorentz force density to dissipation forces similarly to the Hartmann number Ha. Contrary to Ha, this new characteristic number is introduced for the Lorentz force generated by the alternating magnetic field with the modulation frequency of $\omega$. The relation between Lo and Ha numbers is as follows

$$Lo = \frac{\omega R_b (\mu_0 I)^2 \sigma_{el}}{v_{ch} \eta_f}, \tag{4}$$

where $v_{ch}$ is the characteristic flow velocity in the pool and $I$ is electric current inside the inductor.

At $Lo \sim 10^5$, hydrodynamic currents are sensitive even to small variations of the Lorentz force density. Under conditions of laser annealing evaluated in the present work, this number takes values up to $Lo \sim 10^4$. However, it is required to note the value may be significantly overestimated for the specific problem considered in the paper. The value $Lo \sim 10^4$ was obtained from the approximation being used previously by the authors in [6,7] to analyze the problem of an electromagnetically levitated drop where the drop and coil are comparable in size. In the considered case, the inductor diameter can be two orders of magnitude larger than the size of the melt pool. Therefore, the apparent value is close to $Lo \sim 1$.

At the same time, the thermocapillary and ablation effect has a strong impact on hydrodynamic processes in the molten pool after laser melting [17,21]. In this regard, the question arises which effects have the greatest impact: the ablation effect of laser exposure or magnetic fields. The solution to this problem follows from additional numerical simulations carried on in the next section.

## 4. Effect of the Alternating EM Field on the Molten Pool

Experimental assessment of the effect of magnetic fields on solidification requires special measurement techniques. In recent studies [34,35], the X-ray radioscopic method has been used for characterization of the fluid flow and its influence on microstructure under the external magnetic field. In the Ga-25 wt%In alloy, the magnetic field affects the stability and formation of microchannels, which leads to multiple freckle defects. The

obtained experimental results are confirmed by extensive numerical simulations carried out in the same work. It was shown that the magnetic field adds new physical phenomena to the solidification process: (i) electromagnetic damping of the liquid metal motion and (ii) interstitial flow due to thermoelectric magnetohydrodynamics. As a result, Ga-rich plumes migrate along the solidification front that leads to the changes in preferential growth of secondary arms and the formation of segregation channels. Therefore, the effect of electromagnetic field on solidification described in literature was used as a motivation for the present paper. In this connection, theoretical assessment of the effect of an alternating electromagnetic field on convection in a melt pool is developed.

In Figure 6 the geometry of the computation domain is plotted. The subdomain $V_1$ is the surface layer remelted by a laser and $V_2$ is the substrate of the sample. The problem is solved in 3D. The boundary $G_3$ is chosen as the plane of symmetry corresponding with the laser beam trajectory. The numerical model is programmed in the commercial software Comsol Multiphysics 5.6 [31].

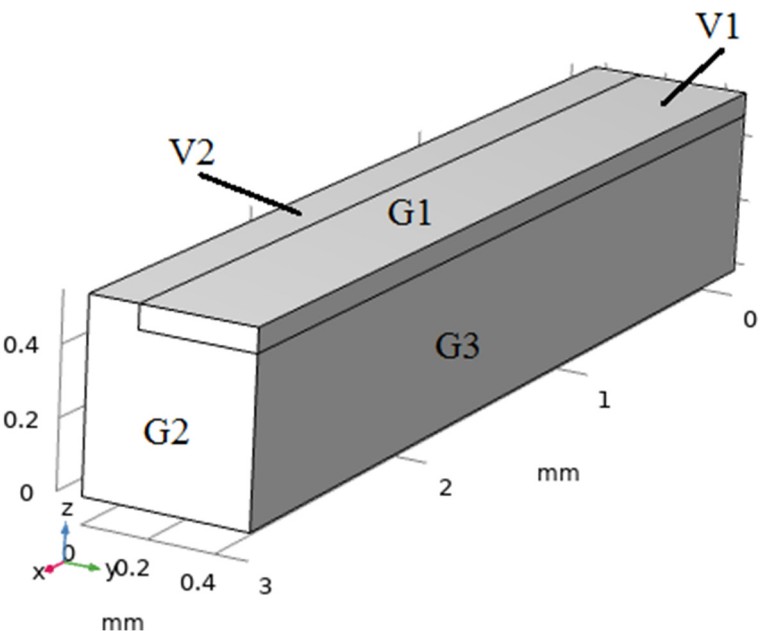

**Figure 6.** Computational domain used for numerical simulation.

The transient temperature distribution in the computational domain is found by numerical simulation of the heat balance equation. The enthalpy of phase transition is introduced in the model for a moving heat source via the effective thermophysical properties dependent on temperature. Such an approach is known in the literature as the enthalpy method. The balance equation is then formulated as

$$\rho C_p(T)\left[\frac{\partial T}{\partial t} + v\cdot\nabla T\right] = \nabla\cdot(k(T)\nabla T), \tag{5}$$

$$\rho = \rho_1(T)\theta_1 + \rho_2(T)\theta_2, \tag{6}$$

where $\rho$ is density; $\theta_1$ and $\rho_1$ are the volume fraction and density of the solid phase, respectively; $\theta_2$ and $\rho_2$ are the volume fraction and density of the liquid phase, respectively; $C_p(T)$ is the effective specific heat dependent on temperature $T$; $k(T)$ is the temperature-dependent thermal conductivity; and $v$ is the flow velocity in the molten pool. The Neumann boundary conditions are accomplished at the boundary $G_1$ as

$$k\frac{\partial T}{\partial n} = q_l + q_{vap} + q_{amb}, \tag{7}$$

where $q_l$ is the heat flux from the laser beam, $q_{vap}$ is the heat flux owing to metal evaporation from the surface and $q_{amb}$ is the heat flux given by convective and radiative cooling in gas atmosphere. At the boundary $G_2$, the condition $n \cdot q = 0$ of thermal isolation is assumed. At $G_3$, the symmetry condition $n \cdot q = 0$ is imposed.

The internal flow in the melt pool is described by the Navier–Stokes equations for incompressible flow in the Boussinesq approximation

$$\rho \left[ \frac{\partial v}{\partial t} + (v \cdot \nabla)v \right] = \nabla \cdot \left[ -pI + \eta \left( \nabla v + \nabla v^T \right) \right] + \rho g + F_L, \tag{8}$$

$$\nabla \cdot v = 0, \tag{9}$$

where $v$ is the flow velocity, $\eta$ is the viscosity, $p$ is pressure, $g$ is the gravitational acceleration and $F_L$ is the Lorentz force density analytically derived by the authors earlier [7]. The calculated intensity of the Lorentz force is depicted in Figure 7 in the cylindrical coordinates $(r, \theta, z)$. The force field has a dominating $z$ component. The Lorentz force is weak in the vicinity of the central zone along to the axis of symmetry. Approaching to the melt–solid interface leads to an increase of the Lorentz force in the $z$ direction. At the same time, the $r$ component appears that facilitates vortex flow. The calculated intensity (Figure 7) shows that its maximum values reach about $\max(F_L) \sim 2500 \,\text{N/m}^3$.

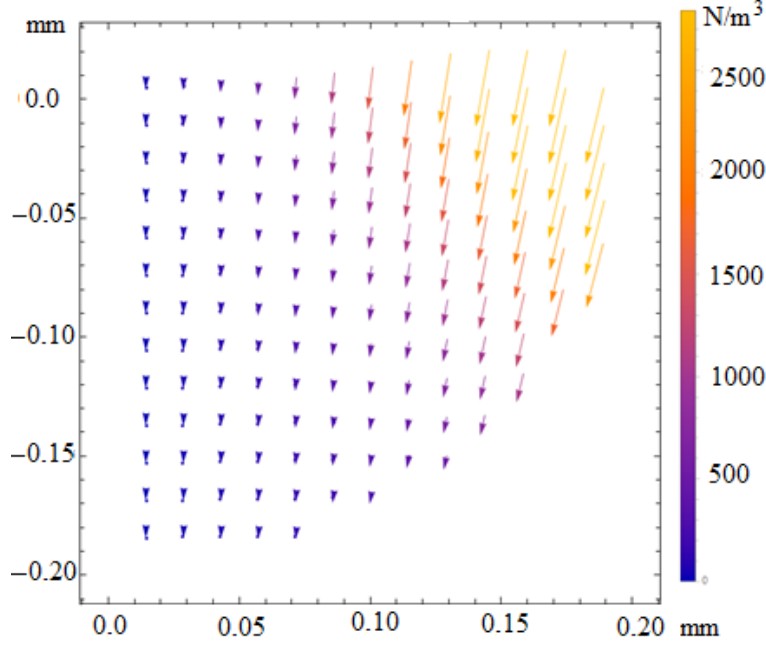

**Figure 7.** Vector distribution of the Lorentz force density in the molten pool formed by the laser beam. Only the right-hand side $z \le 0$, $r \ge 0$, $\sqrt{r^2 + z^2} \le R_b$, $\theta = 0$ of the computational domain is depicted in the cylindrical coordinates due to its rotational symmetry with respect to the Oz axis. The vector length is proportional to the modulus of the Lorentz force density [7].

These data reveal that the Lorentz force is substantially weaker than the gravity force with the intensity of $\rho g \sim 70,000 \,\text{N/m}^3$.

The momentum balance equation in the solid-phase domain $V_1$ is solved under the assumption that the material is a highly viscous fluid, and its viscosity depends on temperature. At the pool's free surface (boundary $G_1$) the vapor pressure, thermocapillary (Marangoni) stress and bubble pressure are properly accounted as

$$\left[ -pI + \eta \left( \nabla v + (\nabla v)^T \right) - \frac{2}{3} \eta (\nabla \cdot v) I \right] n = -p^{recoil} n + \sigma (\nabla_s \cdot n) n + \nabla_s \sigma, \tag{10}$$

where $p^{recoil}$ is the saturation vapor pressure [36], $\sigma$ is the surface tension coefficient and $\nabla_s$ is the spatial derivative along the free surface. The terms $\nabla_s\sigma = \gamma\nabla_s T$ and $\sigma(\nabla_s \cdot n)n$ represent the thermocapillary stress and bubble pressure, respectively. At the $G_2$ boundary and at the bottom boundary, the slip condition is implied. At $G_3$, the symmetry plane of the flow field is set.

The obtained system of equations is discretized by the finite element method. The free surface is modeled by the Arbitrary Lagrangian-Eulerian (ALE) method. The parameters used in simulations are listed in Table 1.

**Table 1.** Thermophysical and processing parameters used in simulations. Data from [37,38].

| Parameter | Symbol | Value | Units |
|---|---|---|---|
| Density of the solid phase | $\rho_1$ | $7950 - 0.50(T - 298[\text{K}])$ | kg/m$^3$ |
| Density of the melt | $\rho_2$ | $6881 - 0.77(T - 1723[\text{K}])$ | kg/m$^3$ |
| Specific heat of the solid phase | $C_{p1}$ | $470 + 0.184(T - 300[\text{K}])$ | J/kg |
| Specific heat of the melt | $C_{p2}$ | 830 | J/kg |
| Thermal conductivity of the solid phase | $k_1$ | $13.4 + 0.0136(T - 300[\text{K}])$ | W/(K m) |
| Thermal conductivity of the melt | $k_2$ | $28.5 + 0.00886(T - 1723[\text{K}])$ | W/(K m) |
| Melting temperature | $T_f$ | 1723 | K |
| Viscosity of the melt | $\eta_f = \nu\rho$ | $5.6 \times 10^{-3}$ | Pa s |
| Viscosity of the solid phase | $\eta_s = \nu\rho$ | $10^5$ | Pa s |
| Thermocapillary coefficient | $\gamma = \frac{d\sigma}{dT}$ | $1.4 \times 10^{-3}$ | $\frac{\text{N}}{\text{m K}}$ |
| Velocity of the laser beam | $v$ | 0.5 | m/s |

The results of numerical simulation of the flow field in the molten pool under conditions of the external alternating magnetic field are shown in Figure 8. At the free surface, the flow is directed towards the pool center in response to the positive thermocapillary coefficient $\gamma = \frac{d\sigma}{dT}$. The flow velocity peaks to 1.92 m/s at the free surface under the external alternating magnetic field. The similar calculation performed in absence of the magnetic field yields the maximum flow velocity of 1.89 m/s.

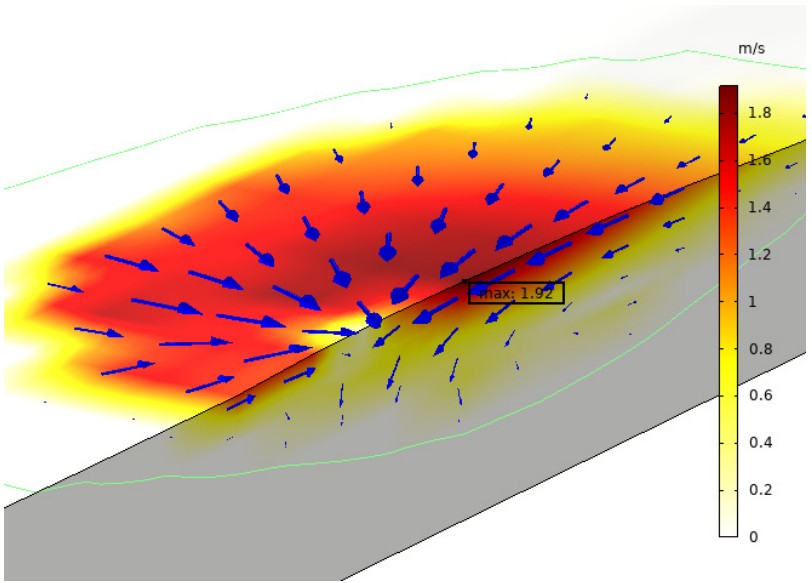

**Figure 8.** Flow field and stream lines in the molten pool calculated under condition of the alternating magnetic field. The green line marks the liquid–solid interface where temperature is equal to the melting temperature. The vector length is proportional to logarithm of the velocity magnitude $\log(|v|)$.

In Figure 9, the distributions of temperature and temperature gradient are simulated under the alternating magnetic field. The calculated maximum temperature at the melt surface attains 2739 K with the temperature gradient $G_T \sim 4.5 \times 10^7$ K/m. The simulations carried without the external magnetic field result in the values 2744 K and $4.0 \times 10^7$ K/m correspondingly. The obtained values evidence that the effect of the alternating magnetic field leads to increase of the temperature gradient between 10 and 15%. The temperature gradient is more sensitive to the external EM field, while convection is less responsive to the magnetic exposure. Therefore, microstructure modification in the annealed stainless steel is possible under condition of its simultaneous laser processing and EM treatment with EMAT. This conclusion follows from the theory of directional solidification [39], where $v$ and $G_T$ are the control parameters that select the microstructure. The experimental confirmation of such an effect found in our study is presented above in the "Experimental Methodology and Microstructure Assessment" Section. Since no changes in the shape of the molten pool were observed, it is assumed that the change in the grain size distribution is caused by the alignment (temporal stabilization) of the velocity field near the solidification front.

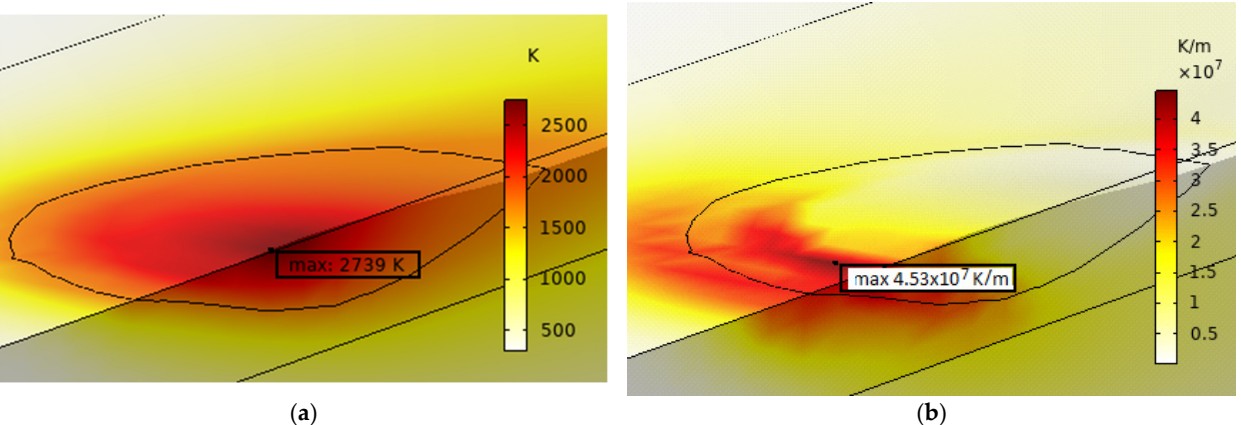

(**a**)　　　　　　　　　　　　　　　　　　　　　　(**b**)

**Figure 9.** Characteristics of the molten pool calculated under condition of the alternating magnetic field: (**a**) temperature field and (**b**) distribution of the temperature gradient.

Based on the performed theoretical and experimental study, one can conclude that the effect of the external alternating field is small in comparison with the effect of laser ultrasonic generation [5]. The EM field dissipates intensively after its interference with the fields generated by eddy currents in the molten pool. If the melt is exposed to an alternating electromagnetic field, Foucault currents are generated in the melt, and these eddy currents are localized in a thin surface layer. The depth of their penetration depends on the frequency of the field and can be calculated by the expression

$$\delta = (\pi \mu_0 \sigma_{el} f)^{-\frac{1}{2}},$$

where f is the frequency of the electromagnetic field and $\mu_0$ is the magnetic permeability. The characteristic depth of decay, i.e., the depth of the electromagnetic skin layer for the considered geometry and current modulation in the coil, is $\delta$ = 1.78 mm. The maximal depth of the melted zone is 0.045 mm, which is few orders of magnitude smaller than the skin depth. Hence, the meaningful effect of the magnetic field can be only expected in a pool of sufficient size. According to the performed analysis, the molten pool should be at least one order of magnitude larger both in the radial and vertical directions than the one registered in SLM processes. In other words, the effect is technologically meaningful if the magnetic field affects the liquid zone of at least 1 mm in size.

## 5. Effect of the Permanent Magnetic Field

The permanent magnetic field may result in various physical phenomena comparing with the alternating field. One can consider the problem where the permanent magnetic field is oriented in the Z direction (Figure 6). The magnetic field density is a variable parameter that takes the values of $B_z = \{0; 0.1; 1; 10\}$ T. The Hartmann number in the considered case is $Ha \approx 48$ at $B_z = 10$ T.

The geometry and mathematical formulation of the problem is similar to the case with the alternating magnetic field. The Lorentz force in Equation (8) is defined by

$$F_L = [j \times B], \tag{11}$$

where $j$ is the current density and $B$ is the magnetic flux density. Then, the current density $j$ is a solution of the following system of equations:

$$j = \sigma_{el}(E + [v \times B]), \tag{12}$$

$$\frac{\partial \rho_q}{\partial t} + div\, j = 0, \tag{13}$$

$$E = -\nabla \varphi, \tag{14}$$

$$\Delta \varphi = -\frac{\rho_q}{\varepsilon_0}, \tag{15}$$

where $\sigma_{el}$ is the electrical conductivity, $E$ is the electric field strength, $\nabla T$ is the temperature gradient, $v$ is the flow velocity in the molten pool, $\varphi$ is the electrical potential and $\rho_q$ is the charge density.

In Figure 10, the results of magnetohydrodynamic simulation in the molten pool under the permanent magnetic field are depicted. The permanent magnetic field results in the deceleration of the hydrodynamic flow. Based on the obtained data, the peak flow velocity decreases from 1.88 m/s at $B_z = 0$ T to 1.82 m/s at $B_z = 10$ T.

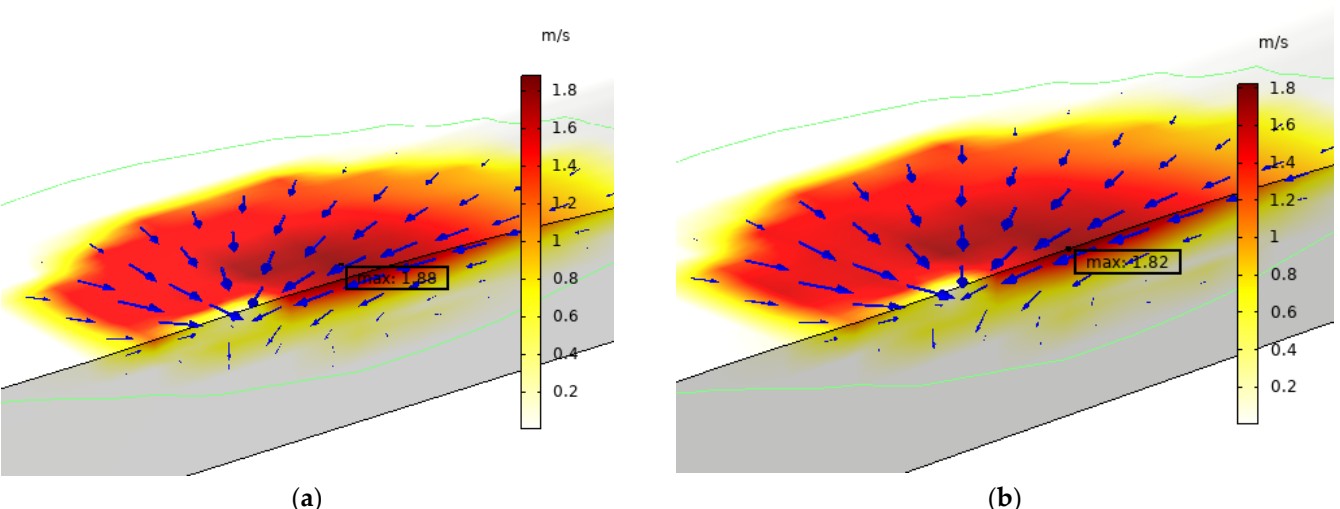

(**a**)                  (**b**)

**Figure 10.** Simulated flow field with depicted directions and velocity magnitude (in m/s) in the molten pool calculated at (**a**) $B_z = 0$ T and (**b**) $B_z = 10$ T under the condition of the permanent magnetic field. The green line marks the liquid–solid interface where the temperature equals the melting temperature. The vector length is proportional to logarithm of the velocity magnitude $\log(|v|)$.

A quantitative comparison of the various flow characteristics at different levels of the magnetic flux density is provided in Table 2. The calculated parameters include (i) the

mean temperature, (ii) time-averaged maximum flow velocity and (iii) spatially averaged flow velocity in the molten pool.

**Table 2.** Cumulative results of a series of numerical simulations at different levels of the magnetic flux density $B_z = 0$, 0.1, 1 and 10 T. The temperature and flow characteristics are averaged in the molten pool.

| Magnetic Flux Density, T | Mean Temperature in the Molten Pool, K | Time-Averaged Maximum Flow Velocity in the Molten Pool, m/s | Spatially Averaged Flow Velocity in the Molten Pool, m/s |
|:---:|:---:|:---:|:---:|
| 0 | 1949 | 1.879 | 0.378 |
| 0.1 | 1949 | 1.878 | 0.378 |
| 1 | 1949 | 1.879 | 0.377 |
| 10 | 1994 | 1.818 | 0.354 |

Analysis of the data in Table 2 shows that the effect of the permanent magnetic field is small or moderate in regards to transport of heat and momentum.

A decrease of the maximum and average flow velocity in the molten pool leads to a decay of the average temperature in the melt. Namely, at $B = 0$ T, the mean flow velocity is 0.378 m/s, and the average temperature is 1949 K. At $B = 10$ T, the same characteristics are 0.354 m/s and 1994 K, respectively.

In Figure 11, the 2D distributions of the temperature field and the temperature gradient along the solidification front are depicted at $B = 0.1$ T. The molten pool is about 0.4 mm in diameter (Figure 11b). As discussed in the previous section, the effect of the external magnetic field is expected at the low level of influence on the thermal conditions at the solidification front. As a result, the computational data at $B = 1$ T and 10 T do not demonstrate any substantial differences in the pool shape (and local solidification velocity) and temperature gradient, which basically control the as-solidified microstructure. For this reason, the results of temperature distribution and temperature gradient along the molten pool interface under the effect of the permanent magnetic field at $B_z = 1$ T and 10 T are not presented.

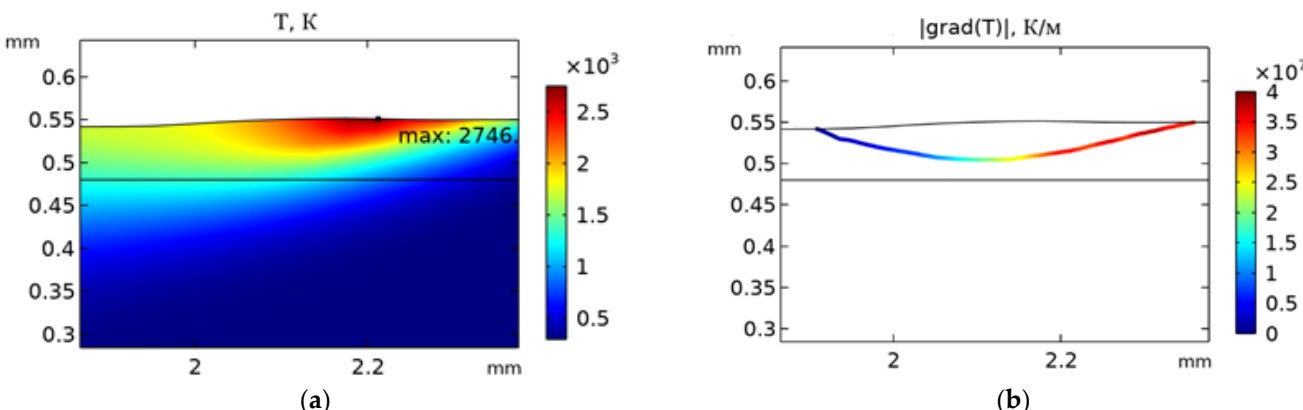

(**a**)                                                                  (**b**)

**Figure 11.** Characteristics of the molten pool calculated under condition of the permanent magnetic field at $B_z = 0.1$ T: (**a**) temperature field and (**b**) distribution of the temperature gradient, where the color line provides the temperature gradient along the solidification front.

## 6. Conclusions

EM stirring is widely used in the metallurgical industry and continuous casting. During the last decade, there has been a number of applied research studies in the field of laser ablation and additive manufacturing. In these processes, such supplementary EM action may change the microstructure selection. In the present paper, an experimental

study of the effect of the alternating EM field on solidification and microstructure has been performed for the AISI 321H stainless steel annealed by laser. The tracked parameter was a distribution of grain areas in the resolidified zone. Modification of the microstructure characteristics was revealed. The area of crystal grains decreases by about 10–15% if extra EM action is applied during laser annealing.

For detailed assessment, theoretical analysis and numerical simulation of the conjugated problem of heat and mass transfer under the effect of alternating and permanent magnetic fields was performed. The molten pool was formed by a laser beam with the effective (80% of the adsorbed power) diameter between 150 and 200 microns. Based on the obtained results, the following conclusions agreed to the registered experimental data are drawn.

(1)     It is confirmed experimentally that the EM field with the frequency of 100 kHz has a minor effect on the microstructure of stainless AISI 321H samples in comparison with ultrasound treatment induced by modulated laser processing.

(2)     The industrial parts produced by additive manufacturing techniques typically have lower plasticity in contrast to casted parts. We have received that additional EM treatment performed simultaneously with laser treatment yields a decrease of the grain's area S (where an estimate of the grain size is related to S as $d \sim \sqrt{S}$) between 10% and 15%. At the same time, the histogram exhibits that the distribution becomes more narrow, i.e., the range of grain areas is smaller. The revealed effect is similar to microstructure homogenization known in heat treatment processing. The revealed new effect can facilitate the improvement of material performance owing to improvement of its plasticity.

(3)     The alternating magnetic field with the frequency up to 100 kHz is characterized by a skin penetration thickness in the range between 1 and 5 mm. Consequently, application of this additional exposure is more effective in laser annealing where the molten pool is sufficiently large and has the size of $L \geq 1$ mm. This explains the physical background in the positive effect of applied electromagnetic fields during welding, which is noted in numerous studies available in the literature.

(4)     The alternating magnetic field results in the formation of the inhomogeneous Lorence force field that has a minor impact on convection in the molten pool in comparison with other physical factors of laser annealing. In the performed laboratory experiments, this effect is registered at the susceptibility threshold. The registered small modification of grain sizes can only be explained by the magnetic field effect, which is substantially mitigated by other physical phenomena in the molten pool.

(5)     According to the literature, the thermoelectrical Seebeck effect provides a meaningful impact on solidification of aluminum alloys owing to the intensification of convection in the zone of dendritic growth. For the case of stainless steel, it has been shown that this effect is small. It happens due to the following factors. First, the sensitivity Seebeck coefficient, which defines generation of thermoelectromotive force, is small for stainless steels. Second, the relaxation time of electromotive force is few orders of magnitude smaller than the time interval required for temperature stabilization in the molten pool.

**Author Contributions:** Conceptualization and writing, M.D.K. and S.L.L.; methodology, E.B.C.; software, G.A.G. and N.N.S.; data curation, A.S.B. and A.A.G.; writing—review and editing, S.A.G.; project administration and methodology, I.A.I. All authors have read and agreed to the published version of the manuscript.

**Funding:** This work was funded by the Grant number MT-96 under the EOTP RosAtom research program. This work was executed within the framework of the State Assignment of the Ministry of Education and Science of the Russian Federation (Projects Nos. BB_2021_121030100003-7 and BB_2021_121030100005-1).

**Informed Consent Statement:** Informed consent was obtained from all subjects involved in the study.

**Conflicts of Interest:** The authors declare no conflict of interest.

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
