# Peer review of "Analysis of the Effect of Magnetic Field on Solidification of Stainless Steel in Laser Surface Processing and Additive Manufacturing"

_metals, doi:10.3390/met12091540_

Round 1
Reviewer 1 Report
The paper discusses the effects of alternating and permanent magnetic fields on the molten zone. Although the paper has novelty in terms of study of magnetic field, significant improvement is needed to consider the paper for publication in Metals.
1、The number of grain sizes smaller than 3 μm(fig.4)in the molten pool is significantly reduced with EMAT on, and this needs to be explained.
2、Why does magnetic fields leads to reduction of the number of large grains.
3、Is there a relationship between the thermocapillary and ablation effect and Lorentz force.
4、Experiments need to be used to verify the accuracy of numerical simulation.
Author Response
Dear Reviewer,
Let us thank you for your time dedicated to reviewing the manuscript and valuable comments and suggestions. In the attached Word file please find the detailed answers on your questions.
We are looking forward to hear from you if the raised questions are properly resolved.
Dr. Svetlana Gruzd
on behalf of all coauthors

Reviewer 2 Report
Thank you for this very interesting manuscript. I think that the considerations around the applicability of magnetic fields to LPBF will be quite useful for the community.
Author Response

(The authors gave the same response as above.)

Reviewer 3 Report
The submitted manuscript is sufficiently attractive for publication. With the following issues solved, it might meet the criteria for publication.
1. How was the external electronmagnetic field added? Any schematic drawing showing the details? The experimental method should be more detailed than the current version.
2. Why the experimental methods and microstructure results in the same section?
3. What exactly are the micrographs in Figure 4, EBSD? Optical Micrographs? SEM?
Author Response

(The authors gave the same response as above.)

Reviewer 4 Report
· My main concern is that both experiments and theoretical analyses showed that the impacts of external magnetic field (either alternating or permanent) on the temperature field, velocity field, and microstructure are not significant. The authors have explained this by the characteristic depth of decay being much larger than the meltpool dimension, and concluded that the molten pool should be at least one order of magnitude larger both in the radial and vertical directions. But this claim is not proved in the present work.
· The text needs to be deeply revised concerning English errors (for example Lines 51, 80, 167, 311) and writing quality. Many sentences are difficult to comprehend.
· Line 95-96: please mention briefly the key findings of this study.
· Some details of the experiments are not clarified:
o Line 99: please address the laser spot diameter, as well
o According to my understanding from lines 100-112, a pulsed mode exposure was employed. If so, please address the pulse frequency, on-time cycle, and resulting long-term power.
o Lines 115-118: please address the amount of plate movement after each laser pulse, and spent time between two laser spots because of this movement.
o What is the inter-pass time for experiments with 5 laser passes?
o How was the EMAT positioned with respect to the plate and to the laser exposure pass?
o EM filed parameters
· Fig. 2: Why is the meltpool depth by the 5 passes exposure shallower than the single-pass one?
· How were the dark areas in Figs. 2 and 3 (which I believe were covered by ultrafine grains) taken into account for the calculation of the histograms in Fig. 4?
· Line 139: at least for the single pass cases, the first characteristic seems not to be correct.
· Line 159-161: this speculation is not clear and convincing. what aspect of EM field? Which dimensionless analysis?
· Please use equation form for the parameters such as Ha (line 163), Lo (line 200), etc.
· Line 211: “However, we did not register any significant impact experimentally”. Which experimental results are being referred to?
· Line 239: density is missing
· Reference/-s are needed for Table 1.
· Line 279: please define ALE (Arbitrary Lagrange-Eulerian?)
· Line 285: “under the external magnetic field” please clarify which type of magnetic field (alternating or constant?)
· Lines 282-300: temperature and fluid velocity fields are compared between two conditions with and without magnetic field, while Figs. 7 and 8 address only one of them. Also, it is not clear in the Fig. 8 captions to which condition it is referred.
· Figs. 7-9: please state clearly in the figure captions which type of magnetic field is applied (alternating, permanent, no magnetic field, …)
· Line 293: increase -> decrease (otherwise, it won't be consistent with the higher velocity peak and the following discussion)
· “The temperature gradient is more sensitive to the external EM field while convection is less responsive to the magnetic exposure.” I don’t understand how the magnetic field can be effective on the thermal gradient without a significant impact on the convection.
· Lines 295-300: I don’t expect the negligible impact on the thermal gradient could result in a considerable microstructure refinement, in particular since no change is expected in the solidification velocity.
· What is the particular highlight found in Fig. 10?
· Conclusions: “The area of crystal grains decreases by about 10–15% if extra EM action is applied during laser annealing” it is not clear what the authors mean by “the area of crystal grains”. Does it mean grain size?
· Conclusions do not summarize the results and discussions in the text. For example, nowhere in the manuscript is plasticity mentioned. However, it is addressed in conclusion 2. Conclusion 3 is not directly proved in this work. Conclusion 1 compares the effects of EM field with the ultrasonic treatment, although such a comparison is not made in this work (only addressed to the literature).
Author Response

(The authors gave the same response as above.)

Round 2
Reviewer 4 Report
I would like to thank the authors for addressing my comments in the revised manuscript. A minor remark remains to be considered. I think that discussing the influence of magnetic fields on microstructures while disregarding dark areas is inaccurate. I would propose utilizing EBSD to explore this area, or just comparing and discussing the dark area fraction in different specimens.
Author Response
Thank you so much for very careful reading of the manuscript which resulted in many helpful comments. These comments ultimately assisted us in improvement of the paper scientific quality. Following your suggestion about the EBSD processing, in principal we agree that such analysis would clear up the questions about the dark region observed in the metallographic specimens. Unfortunately, the authors have difficulties with access to the EBSD equipment. Instead we performed additional evaluation of the samples and added a separate paragraph with discussion of this feature in the text.
